# LEVERAGING BEHAVIORAL CLONING FOR REPRESENTATION ALIGNMENT IN CROSS-DOMAIN POLICY TRANSFER

## ABSTRACT

The limited transferability of learned policies is a major challenge that restricts the applicability of learning-based solutions in decision-making tasks. In this paper, we present a simple method for aligning latent state representations across different domains using unaligned trajectories of proxy tasks. Once the alignment process is completed, policies trained on the shared representation can be transferred to another domain without further interaction. Our key finding is that multi-domain behavioral cloning is a powerful means of shaping a shared latent space. We also observe that the commonly used domain discriminative objective for distribution matching can be overly restrictive, potentially disrupting the latent state structure of each domain. As an alternative, we propose to use maximum mean discrepancy for regularization. Since our method focuses on capturing shared structures, it does not require discovering the exact cross-domain correspondence that existing methods aim for. Furthermore, our approach involves training only a single multi-domain policy, making it easy to extend. We evaluate our method across various domain shifts, including cross-robot and cross-viewpoint settings, and demonstrate that our approach outperforms existing methods that employ adversarial domain translation. We also conduct ablation studies to investigate the effectiveness of each loss component for different domain shifts.

## 1 INTRODUCTION

Humans have an astonishing ability to learn skills in a highly transferable way. Once we learn a route from home to the station, for example, we can get to the destination using various modes of transportation (e.g., walking, cycling, or driving) in different environments (e.g., on a map or in the real world), disregarding irrelevant perturbations (e.g., weather, time, or traffic conditions). We identify the underlying similarities across situations, perceive the world, and accumulate knowledge in our way of abstraction. Such abstract knowledge can be readily employed in diverse similar situations. However, it is not easy for autonomous agents. Agents trained with reinforcement learning (RL) or imitation learning (IL) often struggle to transfer knowledge acquired in a specific situation to another. This is because the learned policies are strongly tied to the representations obtained under a particular training configuration, which is not robust to changes in an agent or an environment.

Previous studies have attempted to address this problem through various approaches. Domain randomization (Tobin et al., 2017; Peng et al., 2018; Andrychowicz et al., 2020) aims to learn a policy that is robust to environmental changes by utilizing multiple training domains. However, it is unable to handle significant domain gaps that go beyond the assumed domain distribution during training, such as drastically different observations or agent morphologies. Numerous methods have been proposed to overcome such domain discrepancies. Earlier approaches learn domain-invariant state representations for imitation using a temporally-aligned dataset across domains (Gupta et al., 2017; Liu et al., 2018b). In cases when we cannot assume such temporal alignment, other approaches utilize an adversarial objective based on domain confusion (Stadie et al., 2017; Yin et al., 2022; Franzmeyer et al., 2022) or cross-domain cycle-consistency (Zakka et al., 2021). These methods require online interaction for adaptation in the target domain to refine a policy, limiting their applicability.

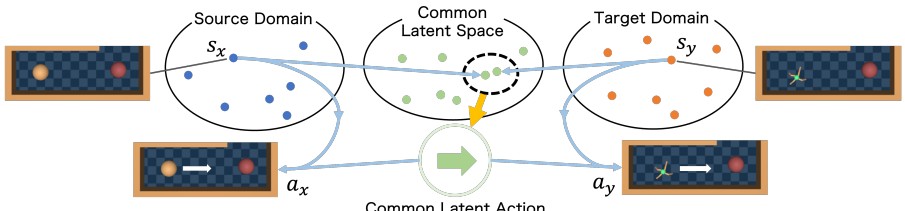

Figure 1: Illustration of a shared representation space. Since semantically similar states are close together in the latent space, we can transfer knowledge across domains through the latent space.

Recently, a few methods have been proposed that do not necessitate online interaction for adaptation (Kim et al., 2020; Zhang et al., 2021; Raychaudhuri et al., 2021). These methods find a cross-domain transformation through the adversarial cross-domain translation of states, actions, or transitions. Although these approaches show promising results, we have found two challenges they face. First, the direct domain translation can be difficult to discover when the discrepancy between domains is not small. For instance, if one agent has no legs while the other agent has multiple legs, we cannot expect a perfect cross-robot translation of information on how the agent walks. Second, these methods rely on signals from adversarial generation and other factors on top of generated elements, lacking a more stable and reliable source of cross-domain alignment.

In this work, we propose a method that does not rely on exact cross-domain correspondence and translation. Our approach learns a shared latent representation across different domains and a common abstract policy on top of it (Figure 1). After achieving the alignment, we can update a common policy for the target task using any learning algorithm with the mappings between the shared space and the original domains frozen. Combined with the frozen mappings, we can readily deploy the learned common policy in either domain without further online interaction. Similar to previous studies (Kim et al., 2020; Zhang et al., 2021), we assume access to a dataset of expert demonstrations of proxy tasks, which are relatively simple tasks used for aligning representation. In contrast to existing methods that stand on adversarial generation with domain confusion, our approach leverages multi-domain behavioral cloning (BC) on proxy tasks as a core component for shaping a shared representation space. We then add a few regularization terms on the latent state distributions to encourage cross-domain alignment. Although adversarial generation with a domain classifier is a commonly used technique to match multiple distributions, we observe that exact matching of distributions is overly demanding and sometimes disrupts the structure of a shared representation space. We instead employ maximum mean discrepancy (MMD) (Gretton et al., 2012), a widely utilized technique in domain adaptation (Long et al., 2013; Tzeng et al., 2014; Baktashmotlagh et al., 2016). We empirically confirm that it has a less detrimental impact on the representation structure. We can optionally add more regularizations on the representation depending on proxy tasks. As an example, we add Temporal Cycle-Consistency learning (TCC) (Dwibedi et al., 2019) to promote state-to-state alignment using temporal information within a task rather than distribution overlap. It is worth noting that our method only requires a policy network and a few loss terms, whereas other methods of offline cross-domain transfer usually require more models and objectives to optimize. This allows us to easily extend our method for better alignment in diverse situations.

We evaluate our approach under various domain shifts, including changes in observation, action, viewpoint, and agent morphology. Our approach outperforms existing methods, particularly when exact domain translation of states or actions is hard to discover. Moreover, our approach demonstrates superior adaptation capabilities to out-of-distribution tasks. We also confirm that MMD performs better than discriminative training in these cases. Additionally, we conduct extensive ablations to investigate the role of each loss term. Perhaps surprisingly, our method shows some capability of cross-domain transfer only with the BC loss when the target task has a similarity to proxy tasks. This is an indication of implicit representation alignment of multi-domain BC.

In summary, our main contributions are as follows:

- We propose a method for cross-domain transfer that acquires a domain-shared feature space leveraging signals from multi-domain imitation in addition to domain confusion regularization with MMD, in contrast to the latest methods that rely on domain translation.

- We experimentally show the efficacy of our method under various domain shifts. Our method outperforms existing methods, especially in cross-robot transfer or cross-viewpoint transfer, where exact domain translation is hard to discover.

- We perform ablations to investigate the effect of each loss component for different domain shifts. We also confirm that the MMD regularization performs better than the domain discriminative loss when the structure of latent states does not differ much between domains.

## 2 Related Work

**Cross-Domain Policy Transfer between MDPs** Transferring a learned policy to a different environment is a long-standing challenge in policy learning. Most of the previous methods acquire some cross-domain metric to optimize and train a policy for a target task using a standard RL algorithm (Gupta et al., 2017; Liu et al., 2018b; 2020; Zakka et al., 2021; Fickinger et al., 2022) or a Generative Adversarial Imitation Learning (GAIL) (Ho & Ermon, 2016)-based approach (Stadie et al., 2017; Yin et al., 2022; Franzmeyer et al., 2022) through online interaction with the environment. An adversarial objective based on domain confusion is typically used to match the state distribution of multiple domains. In the reward calculation for standard RL, the distance between temporally-corresponding states (Gupta et al., 2017; Liu et al., 2018b) or the distance from the goal (Zakka et al., 2021) in the latent space is often used. Similar to our method, some recent approaches do not assume online interaction for the adaptation to the target task. Kim et al. (2020); Zhang et al. (2021); Raychaudhuri et al. (2021) learn mappings between domains by adversarial generation of transitions or by CycleGAN (Zhu et al., 2017), while Zhang et al. (2020) impose domain confusion on its state representation to address domain shift in observation. Our approach predicts actions without learning cross-domain mappings and focuses only on the shared structure, and also utilizes multi-domain BC for the representation alignment. In the experiment, we show that our approach can handle larger and more diverse domain shifts in spite of the simplicity compared to other baselines. For cross-robot transfer, Hejna et al. (2020) train a portable high-level policy by using a subgoal position as a cross-robot feature. Gupta et al. (2022) cover a morphology distribution to generalize to unseen robots. We intend to perform direct policy transfer without making domain-specific assumptions.

**State Abstraction for Transfer** Theoretical aspects of latent state representation have been analyzed in previous studies. There exist several principled methods of state representation learning for transfer such as bisimulation (Castro & Precup, 2010) and successor features (Barreto et al., 2017). Recently, Gelada et al. (2019) proved that the quality of a value function is guaranteed if the representation is sufficient to predict the reward and dynamics of the original Markov decision process (MDP). In a similar context, Zhang et al. (2020); Sun et al. (2022) provide performance guarantees in multi-task settings or cross-domain transfer.

**Unsupervised Domain Adaptation & Correspondence Learning** Domain adaptation with unaligned datasets has been intensively studied in computer vision. Domain confusion is widely used to match the distributions of multiple domains (Tzeng et al., 2014; Ganin et al., 2016; Tzeng et al., 2017). CycleGAN (Zhu et al., 2017) finds a cross-domain translation by generating the corresponding instances in another domain. If the output space is consistent between domains, we can enforce invariance on downstream components before and after the translation (Hoffman et al., 2018; Rao et al., 2020). Additionally, temporal relationships between frames (Sermanet et al., 2018; Dwibedi et al., 2019), cycle-consistency in agent trajectories (Zhang et al., 2021; Wang et al., 2022), and optimal transport methods (Fickinger et al., 2022) can be exploited to acquire domain translation or domain-invariant features. These features can subsequently be used for reward shaping in cross-domain imitation (Zakka et al., 2021).

## 3 Problem Formulation

We consider a Markov decision process (MDP): $\mathcal{M} = (\mathcal{S}, \mathcal{A}, R, T)$, where $\mathcal{S}$ is a state space, $\mathcal{A}$ is an action space, $R : \mathcal{S} \times \mathcal{A} \to \mathbb{R}$ is a reward function, and $T : \mathcal{S} \times \mathcal{A} \times \mathcal{S} \to \mathbb{R}_{\geq 0}$ is a transition function. We also define domain $d$ as a tuple $(\mathcal{S}_d, \mathcal{A}_d, T_d)$ and denote an MDP in domain $d$ as $\mathcal{M}_d : (\mathcal{S}_d, \mathcal{A}_d, R_d, T_d)$. The aim of this paper is to transfer knowledge of a source MDP $\mathcal{M}_x$ in a source domain $x$ to a target MDP $\mathcal{M}_y$ in a target domain $y$. Here we assume that these MDPs

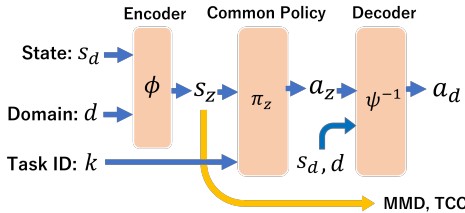 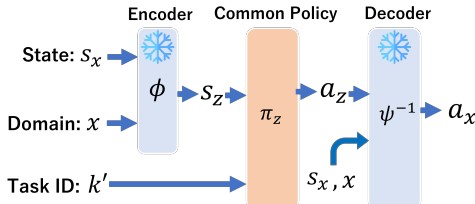

(a) Alignment phase. All modules are trainable.  (b) Adaptation phase. Only common policy is updated.

Figure 2: Overview of the training and inference procedure of our method. (a) In the alignment phase, we train all modules of the policy via BC and regularization terms using trajectories of proxy tasks to obtain cross-domain representation alignment. (b) In the adaptation phase, we only update the common policy to adapt to the target task in the source domain. In inference, we can use the updated policy combined with the encoder and decoder already trained in the alignment phase.

share a common latent structure which is also an MDP $\mathcal{M}_z$. Formally, we assume the existence of state mappings $\phi_x : \mathcal{S}_x \to \mathcal{S}_z, \phi_y : \mathcal{S}_y \to \mathcal{S}_z$ and action mappings $\psi_x : \mathcal{A}_x \to \mathcal{A}_z$, $\psi_y : \mathcal{A}_y \to \mathcal{A}_z$ which translate states $s_x, s_y$ or actions $a_x, a_y$ into shared states $s_z$ or actions $a_z$, respectively, satisfying $T_z(\phi_d(s_d), \psi_d(a_d), \phi_d(s'_d)) = T_d(s_d, a_d, s'_d)$ and $R_z(\phi_d(s_d), \psi_d(a_d)) = R_d(s_d, a_d)$ for all $s_d, a_d, s'_d$ in each domain $d$. In short, we assume that the common latent MDP is expressive enough to reproduce the dynamics and reward structure of both MDPs.

Our goal is to learn the state mapping functions $\phi_x, \phi_y$ and the action mapping function $\psi_x, \psi_y$ so that any policy learned in the common latent space $\pi_z(a_z|s_z) : \mathcal{S}_z \times \mathcal{A}_z \to \mathbb{R}_{\geq 0}$ can be immediately used in either MDP combined with the obtained mappings. In this paper, we use a deterministic policy and denote the latent policy as $\pi_z(s_z) : \mathcal{S}_z \to \mathcal{A}_z$, although we can easily extend it to a stochastic policy. We learn these mappings using expert demonstrations of *proxy tasks* $\mathcal{K}$, which are simple tasks where we can easily collect demonstrations: $\mathcal{D} = \{(\mathcal{D}_{x,k}, \mathcal{D}_{y,k})\}_{k=1}^{|\mathcal{K}|}$, where $\mathcal{D}_{d,k} = \{\tau_{d,k,i}\}_{i=1}^N$ is a dataset of $N$ state-action trajectories $\tau_{d,k,i}$ of an expert in domain $d$, task $k$. After we learn the relationships, we update the policy for a novel target task $k' \notin \mathcal{K}$ in the source domain $x$, and finally evaluate its performance in the target domain $y$.

## 4 LEARNING COMMON POLICY VIA REPRESENTATION ALIGNMENT

In this work, we aim to learn state mapping functions $\phi_x, \phi_y$, and action mapping functions $\psi_x, \psi_y$ or equivalents, and use them to transfer the policy learned in one domain to another. Our algorithm consists of two steps as illustrated in Figure 2: (i) Cross-domain representation alignment, (ii) Policy adaptation to a target task in the source domain. We call them the *alignment* phase and the *adaptation* phase, respectively. After the adaptation phase, the learned policy of a target task can be readily used in the target domain without any fine-tuning or further interaction with the target domain (Gupta et al., 2017; Liu et al., 2018b; Zakka et al., 2021; Fickinger et al., 2022; Yin et al., 2022; Franzmeyer et al., 2022), or a policy learning in the mapped target domain (Raychaudhuri et al., 2021).

### 4.1 CROSS-DOMAIN REPRESENTATION ALIGNMENT

In the alignment phase, we aim to learn the state and action mappings and acquire a domain-shared feature space that can be used in either the source domain or the target domain. We represent our policy as a simple feed-forward neural network as shown in Figure 2. It consists of three components: a state encoder, a common policy, and an action decoder. They correspond to $\phi(s)$, $\pi(s_z)$, and $\psi^{-1}(a_z)$, respectively. $\psi^{-1}$ additionally takes a raw state $s$ to replenish the domain-specific information for the action prediction in a domain. Note that we feed domain ID $d$ to the encoder and the decoder, and one-hot encoded task ID $k$ to the common policy, instead of using separate networks for each domain, to handle multiple domains and tasks with a single model.

We train the network via multi-domain BC with two regularization terms: MMD (Gretton et al., 2012) loss and TCC (Dwibedi et al., 2019). Each objective plays a different role in shaping the aligned representation space. Through BC, the model learns the relationship between raw states and

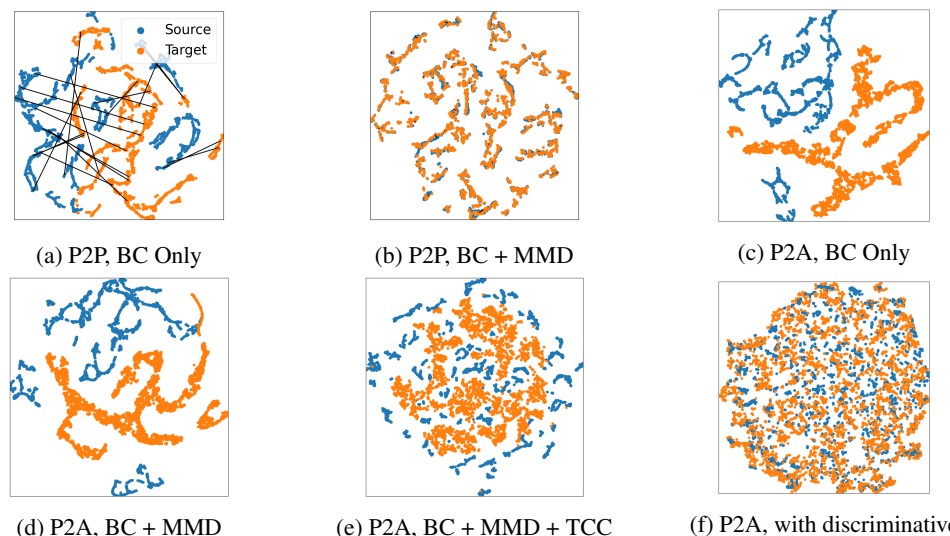

(a) P2P, BC Only      (b) P2P, BC + MMD      (c) P2A, BC Only

(d) P2A, BC + MMD      (e) P2A, BC + MMD + TCC      (f) P2A, with discriminative

Figure 3: Effect of each objective on the latent state distribution $(s_z)$. The representations are projected to 2D space by t-SNE (Van der Maaten & Hinton, 2008). Here we sample corresponding states from two domains of the Maze environment in our experiment (c.f. Section 5.1). (a-b) A plot for P2P-medium environment, where the two domains are the same except for the format of states and actions. Black lines connect 20 corresponding state pairs. BC shapes the latent space in each domain separately (3a), and the MMD loss encourages the distribution alignment. (3b). (c-f) A plot for P2A-medium environment, where two domains have a large discrepancy. With the help of MMD and TCC, our method obtains a shared latent space keeping the original structure (3e). If we use discriminative loss instead of MMD, the latent structure can be disrupted (3f).

corresponding expert actions. We simply calculate the L2 loss on raw actions as follows:

$$\mathcal{L}_{\text{BC}} = \mathbb{E}_{(s_d, a_d, d, k) \sim \mathcal{D}} \left[ \|\psi_d^{-1}(\pi_z(\phi_d(s_d), k), s_d) - a_d\|^2 \right]. \tag{1}$$

If we naïvely optimize this objective with trajectories from both domains, the model can learn a separate state representation in each domain. If temporally-aligned trajectories are available (i.e. $(\phi_x(s_x^t), \psi_x(a_x^t)) = (\phi_y(s_y^t), \psi_y(a_y^t))$ at timestep $t$), we can directly make corresponding representations $\phi_x(s_x), \phi_y(s_y)$ or $\psi_x(a_x), \psi_y(a_y)$ close together (Gupta et al., 2017). However, we do not assume such an alignment and thus need to use other regularizations to align the representations.

MMD is a non-parametric metric that compares the discrepancy between two distributions based on two sets of data points. Optimizing the MMD is widely used in domain adaptation in computer vision (Long et al., 2013; Tzeng et al., 2014; Baktashmotlagh et al., 2016), and is also used to align the support of two action distributions (Kumar et al., 2019). We apply it to latent states encoded in the source domain and target domain:

$$\mathcal{L}_{\text{MMD}} = \mathbb{E}_{s_x, s_x' \sim \mathcal{D}_x}[f(\phi_x(s_x), \phi_x(s_x'))] + \mathbb{E}_{s_y, s_y' \sim \mathcal{D}_y}[f(\phi_y(s_y), \phi_y(s_y'))]$$
$$- 2\mathbb{E}_{s_x \sim \mathcal{D}_x, s_y \sim \mathcal{D}_y}[f(\phi_x(s_x), \phi_y(s_y))], \tag{2}$$

where $\mathcal{D}_d$ is a dataset of domain $d$ and $f$ is a kernel function that measures the similarity between sets of data points. We adopt the Gaussian kernel as $f$, combined with batch-wise distance normalization to avoid representation corruption. See Appendix C.3 for the details. For a similar purpose, domain discriminative training, where we train state encoder or domain translation functions to fool the domain discriminator, is frequently used in the context of cross-domain policy transfer (Kim et al., 2020; Zhang et al., 2020; Raychaudhuri et al., 2021; Franzmeyer et al., 2022). However, this objective enforces the complete match of the distributions even if the structure of latent space or state frequency differs. Figure 3e and 3f show how the MMD and discriminative objective affect the alignment in such a case. MMD encourages the distribution overlap modestly, whereas discriminative training forces the exact match disregarding the original structure. We also observed that, in visual input settings, aligning representations containing image embedding with the MMD is more stable than with discriminative training. We evaluate the effect of this difference in our experiment.

We additionally impose temporal alignment regularization on the representation $s_z$ using TCC. Given two trajectories of the same task, TCC enforces cycle consistency between corresponding frames in the latent space. TCC only selects the frame, in contrast to CycleGAN (Zhu et al., 2017) generating the counterpart, which makes the learning easier. Specifically, we randomly sample two state trajectories of the same task from both domains and encode each frame with the state encoder $\phi_d(s_d)$. Here we denote the two encoded trajectories as $U = (u_1, \cdots u_{|U|})$, $V = (v_1, \cdots v_{|V|})$, where $u_t = \phi_{d_1}(s_{d_1}^t)$, $v_t = \phi_{d_2}(s_{d_2}^t)$. For each state $u_i$ in $U$, we calculate the soft nearest-neighbor in $V$: $\tilde{v}^i = \sum_j^{|V|} \text{softmax}_j(-\|u_i - v_j\|) \cdot v_j$. Then we choose a state in the first trajectory that is closest to $\tilde{v}^i$, and optimize the cross-entropy loss so that this sequence of mappings comes back to the original frame $u_i$. The final TCC objective is $\mathcal{L}_{\text{TCC}} = -\sum_i^{|U|} \sum_k^{|U|} \mathbf{1}_{k=i} \log(y_k^i)$, where $y_k^i = \text{softmax}_k(-\|\tilde{v}^i - u_k\|^2)$, and $\mathbf{1}$ is an indicator function. Note that, as TCC requires unique correspondences for each state, proxy tasks should be designed to have no repetition in a trajectory.

Combining these three objectives, (1), (2), and $\mathcal{L}_{\text{TCC}}$, we have our objective for the alignment phase:

$$\min_{\phi, \pi_z, \psi^{-1}} \mathcal{L}_{\text{align}} = \min_{\phi, \pi_z, \psi^{-1}} \mathcal{L}_{\text{BC}} + \lambda_{\text{MMD}} \mathcal{L}_{\text{MMD}} + \lambda_{\text{TCC}} \mathcal{L}_{\text{TCC}}, \tag{3}$$

where $\lambda_{\text{MMD}}$ and $\lambda_{\text{TCC}}$ are hyperparameters that define the importance of the regularization terms. Figure 3 shows the effect of each loss term in our experiment. We discuss it more in Section 5.3.

In some experimental setups, we observe that the state input for the action decoder can degrade the performance because the decoder can obtain all necessary information except task ID $k$ without the common policy. We can eliminate this effect by removing the state input for the decoder when the action prediction does not require domain-specific state information.

## 4.2 POLICY ADAPTATION

In the adaptation phase, we update the common policy on top of the aligned latent space trained in the alignment phase. This adaptation can be solely in the source domain with any learning algorithm including reinforcement learning as long as the latent space is fixed. As described in Figure 2, we freeze the weights of the encoder and decoder during the process. In our experiments, we update the common policy by BC using expert trajectories in the source domain $\mathcal{D}_{x,k'}$:

$$\min_{\pi_z} \mathcal{L}_{\text{adapt}} = \min_{\pi_z} \mathcal{L}_{\text{BC}}. \tag{4}$$

When the discrepancy between domains is not small, or the alignment is imperfect, the update only with source domain data can make a common policy more or less domain-specific. This issue can be addressed by mixing the data used in the alignment phase.

## 5 EXPERIMENTS

We conduct experiments to answer the following questions: (i) Can our method align latent states of different domains? (ii) Can our method achieve zero-shot cross-domain transfer across various settings? (iii) How does each loss contribute to transfer under different types of domain shifts?

## 5.1 ENVIRONMENTS AND TASKS

For locomotion tasks, we use the Maze environment of D4RL (Fu et al., 2020) (Figure 4). An agent explores two types of mazes, *umaze* and *medium*, toward a designated goal position. An agent observes proprioceptive state vectors. A task is defined as a combination of a starting area and a 2D position of a goal. Tasks with different goals from the goal of a target task are used as proxy tasks. In *OOD* (Out-of-Distribution) variation, the target task is to take a detour as performed in given source domain demonstrations (Figure 4f), despite the alignment dataset only containing trajectories that take the shortest path. We create two setups with different domain shifts. (i) Point-to-Point (**P2P**): a Point agent learns from another Point agent with different observation and action spaces. The source domain and target domain are essentially identical and we can calculate ground-truth correspondence between domains for evaluation. For ablation, we create *P2P-obs* for the medium maze, where we only keep the observation shift. (ii) Point-to-Ant (**P2A**): an Ant agent learns from a Point agent, which has drastically different observations, actions, dynamics, and physical capabilities.

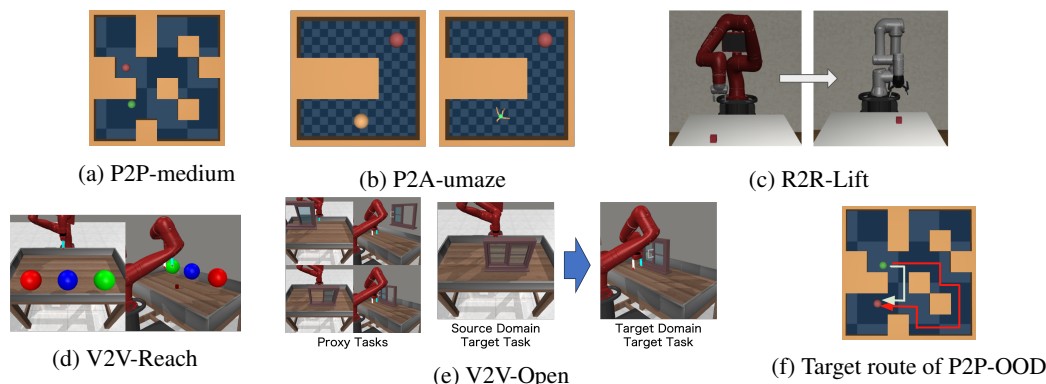

(a) P2P-medium
(b) P2A-umaze
(c) R2R-Lift

(d) V2V-Reach
(e) V2V-Open
(f) Target route of P2P-OOD

Figure 4: (a-e) Pictures of P2P, P2A, R2R-Lift, and V2V. The red points in the mazes (4a) and (4b) show the goals, which are not observable for the agents. (f) The route for OOD tasks. The red arrow shows the target route, while the dataset only contains the shortest paths as the green arrow.

For manipulation, we create three tasks. (i) Robot-to-Robot Lift task (**R2R-Lift**) from robosuite (Zhu et al., 2020): a robot has to learn from another robot with a different number of joints and different grippers to pick up a target block in a position unseen during the alignment phase (Figure 4c). The observations are low-dimensional vectors with up to 37 dimensions. Both robots are controlled by delta values of a 3D position of the end effector and a 1D gripper state, although the outcome of the action can vary to some extent. A single goal position in 3D space is selected as a target task and other goals that are not in the same height or same 2D position are used for proxy tasks. In Viewpoint-to-Viewpoint (V2V) environments constructed based on environments in Meta-World (Yu et al., 2019), a robot learns from demonstrations from a different viewpoint. The robot observes an RGB image from a specific viewpoint in addition to common proprioceptive inputs. We use two setups. (ii) **V2V-Reach**: the robot needs to move its arm to a goal shown as a ball with a specific color in an image. The order of balls is randomly initialized. We use a single color for a target task and use the rest for proxy tasks. (iii) **V2V-Open**: the robot needs to open the window in a random position. The proxy tasks only contain trajectories of closing the window, where the robot moves its arm in the opposite direction. For the details, please refer to Appendix C.1.

## 5.2 BASELINES

We refer to our method as **PLP** (**P**ortable **L**atent **P**olicy) and compare PLP with the following approaches. We also create *PLP-disc* where we replace our MMD loss with the domain discriminative loss. For the reason mentioned in Section 4.1, we do not provide states for the decoder in R2R and V2V. **GAMA** (Kim et al., 2020) learns direct cross-domain mappings of states and actions via adversarial training on generated transitions using a dynamics model. GAMA solves the target task using the updated source domain policy combined with the learned cross-domain translation. **CDIL** (Raychaudhuri et al., 2021) learns a state translation function with CycleGAN (Zhu et al., 2017) for cross-domain transfer. CDIL additionally employs information on task progression via regression of the progression, which has a similar role to that of TCC. **Contextual policy** (*Contextual* for short) is a policy with Transformer (Vaswani et al., 2017) architecture that takes source domain demonstration in the encoder and takes the observation history in the decoder, and outputs the next action. It only has the alignment phase as it requires a pair of demonstrations from both domains for the training. **BC** learns a flat policy that digests a state, domain ID, and task ID at once. It is trained with the same parameters as PLP. See Appendix C.3 and C.4 for the details.

## 5.3 ALIGNMENT QUALITY

**Quantitative Evaluation** We evaluate the quality of the alignment in P2P, where we know the ground-truth state correspondence between domains. As the two domains are essentially identical, we expect that corresponding states are mapped to the same latent representation. For evaluation, we sample 1000 corresponding state pairs from both domains, encode them into the latent space, and test if we can find corresponding states based on the distance in the latent space. We compare PLP with

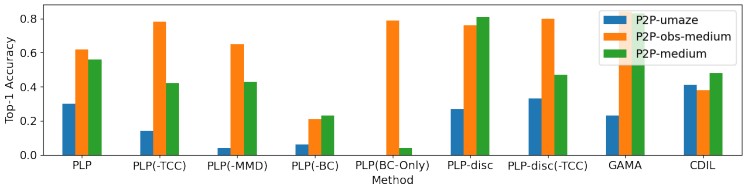

Figure 5: Alignment scores. The values are top-1 accuracies of finding a corresponding state from 1k random states based on the distance in the latent space (or state space for GAMA).

Table 1: The success rates averaged over nine runs. In P2A-OOD, no method completes the task.

| Task | PLP (Ours) | GAMA | CDIL | Contextual | BC |
|------|-----------|------|------|-----------|-----|
| P2P-umaze | $0.68 \pm 0.39$ | $0.71 \pm 0.40$ | $\mathbf{1.00 \pm 0.00}$ | $0.47 \pm 0.08$ | $0.28 \pm 0.23$ |
| P2P-medium | $\mathbf{0.84 \pm 0.08}$ | $0.42 \pm 0.19$ | $0.62 \pm 0.35$ | $0.28 \pm 0.20$ | $0.24 \pm 0.22$ |
| P2P-OOD | $\mathbf{0.60 \pm 0.46}$ | $0.31 \pm 0.38$ | $0.23 \pm 0.37$ | $0.00 \pm 0.00$ | $0.00 \pm 0.00$ |
| P2A-umaze | $\mathbf{0.50 \pm 0.41}$ | $0.01 \pm 0.03$ | $0.01 \pm 0.02$ | $0.32 \pm 0.25$ | $0.33 \pm 0.43$ |
| P2A-medium | $\mathbf{0.70 \pm 0.17}$ | $0.00 \pm 0.01$ | $0.00 \pm 0.00$ | $0.00 \pm 0.00$ | $0.17 \pm 0.12$ |
| R2R-Lift | $\mathbf{0.71 \pm 0.21}$ | $0.09 \pm 0.15$ | $0.00 \pm 0.00$ | $0.09 \pm 0.19$ | $0.22 \pm 0.40$ |
| V2V-Reach | $\mathbf{0.68 \pm 0.24}$ | $0.12 \pm 0.18$ | $0.03 \pm 0.05$ | $0.14 \pm 0.12$ | $0.11 \pm 0.05$ |
| V2V-Open | $\mathbf{0.68 \pm 0.14}$ | $0.07 \pm 0.01$ | $0.00 \pm 0.00$ | $0.00 \pm 0.01$ | $0.00 \pm 0.00$ |

the ablated variants of PLP and the baselines with respect to top-1 accuracy. The results in Figure 5 show that the regularization terms of PLP enhances the quality of the alignment, although TCC sometimes does not improve the performance. We observe that BC loss helps the alignment more than MMD and TCC do in PLP training. It is also worth mentioning that, in P2P-obs, where there is no action shift, representations are well aligned only with the BC loss. PLP achieves comparable performance to existing methods. PLP-disc and GAMA perform even better, but the better score does not necessarily lead to good transfer performance (Section 5.4).

**Qualitative Evaluation** We visualize the latent state distributions for Maze experiments in Figure 3. In P2P, the MMD loss effectively aligns the representations. In P2A, where the domains differ significantly due to varying agent morphologies, neither MMD loss nor TCC alone makes a difference we can visually confirm, and we need both to get them closer. The MMD loss moves distributions toward the same space while preserving their structure. In contrast, with the discriminative loss, distributions overlap but seem to lose the original structure in each domain, resulting in the performance drop in Table 1 in the next section. This indicates that distribution matching can be harmful unless it keeps exact correspondence. For R2R and V2V-Reach, we present interpretable cross-domain correspondence on the plots in Appendix B.1.

## 5.4 Cross-Domain Transfer Performance

Table 1 summarizes the success rate of a target task in each setting. PLP outperforms the baselines in most settings ranging from cross-morphology transfer to cross-viewpoint transfer. GAMA and CDIL show very limited performance in P2A and manipulation tasks including visual-input environments, where cross-domain translation that these methods rely on is not trivial. PLP instead reduces each MDP to the common one and avoids this issue. PLP also utilizes signals from joint multi-domain imitation and does not fully rely on unstable adversarial training. Contextual policy struggles to adapt to OOD tasks including V2V. It is because Contextual could only see proxy tasks and does not have a way to adapt to unseen tasks, as it needs data from both domains to update the policy. Contextual also shows suboptimal performance in P2A and R2R, where an agent needs precise control. We observe that the control by Contextual is less precise than PLP possibly due to the lack of adaptation phase to unseen target tasks. BC shows transferring capability to some extent but the performance is suboptimal because it updates all parameters at adaptation even if it obtains certain implicit alignment via imitation. It also completely fails at highly OOD tasks as it learns OOD target tasks only on the latent space of a source domain when distributions of domains are separate. We provide the visualization of agents in the tasks in B.2.

Table 2: Comparison of the success rates between ablated variants of PLP. The standard deviations are omitted for space restrictions. The full table is available in Table 3 in Appendix.

| Task | PLP-full | PLP-disc | BC + MMD | BC + disc | BC + TCC | BC Only |
|---|---|---|---|---|---|---|
| P2P-obs-medium | 0.92 | 0.90 | **0.94** | 0.91 | 0.87 | 0.92 |
| P2P-medium | 0.84 | 0.88 | **0.93** | 0.81 | 0.72 | 0.48 |
| P2A-medium | **0.70** | 0.52 | 0.54 | 0.45 | 0.62 | 0.52 |
| R2R-Lift | **0.71** | 0.39 | 0.63 | 0.11 | 0.54 | 0.64 |
| V2V-Reach | 0.68 | 0.68 | **0.70** | 0.37 | 0.43 | 0.43 |
| V2V-Open | **0.68** | 0.33 | 0.61 | 0.00 | 0.25 | 0.00 |

## 5.5 Effect of Each Loss Term on Transfer Performance

To investigate the impact of each loss term on transfer performance, we remove terms from the objective or replace the MMD loss with the domain discriminative loss. Table 2 presents the success rates. We confirm that the MMD effectively enhances the performance of PLP. While TCC aids alignment, the improvement from BC+MMD is relatively small and task-dependent. The contrast between the BC baseline and BC-only also highlights the advantages of the training strategy of PLP.

Although PLP-disc performs as well as PLP does in P2P, the discriminative term does not benefit the performance, and can even deteriorate the performance of BC+TCC in P2A and manipulation tasks, where the source domain and target domain are not identical (i.e., have differences in something other than format). The cause of this difference seems to be that the discriminative objective is excessively strong that the representation space cannot retain the original structure necessary for precise action prediction when the state distributions of domains are not the same in the first place. MMD loss avoids this issue and PLP achieves better performance than BC+TCC. Besides, in V2V tasks, we find that the alignment with the discriminative objective is unstable and fails at overlapping distributions in contrast to the performance in non-visual observation environments.

In environments with consistent action format and less-OOD target tasks such as P2P-obs and R2R, BC-only performs similarly to PLP-full. In V2V environments, which do not have action shifts either, the performance of BC-only decreases as the target task is getting OOD. These results indicate that multi-domain BC seems to provide useful signals for representation alignment. As mentioned in Section 5.3, BC-only achieves nearly perfect alignment in P2P-obs, and also, the alignment possibly occurs in another place than where we impose the constraints as well. This good performance aligns with the recent success in large-scale imitation across multiple domains and robots with a shared architecture and action format (Jang et al., 2021; Ebert et al., 2022; Brohan et al., 2022). For additional discussions on the scaling of dataset, the number of domains and proxy tasks, hyperparameter sensitivity, and learning from state-only demonstrations, please refer to Appendix A.

## 6 Conclusion

In this study, we introduce PLP, a novel method for learning a domain-shared policy for cross-domain policy transfer. PLP leverages multi-domain BC for cross-domain representation alignment in addition to MMD loss and TCC and avoids cross-domain translation that is difficult to learn. Our experimental results show the effectiveness of PLP across situations such as cross-robot, cross-viewpoint, and OOD-task settings. Interestingly, our results indicate that multi-domain BC sometimes implicitly aligns the representation when there is no large domain shift in action space. We also confirm that MMD loss helps to align the latent distributions keeping their original structure, whereas domain discriminative training can disrupt them when it forces complete overlap.

Although PLP shows promising results, it has several limitations. The performance of PLP is not stable and it fails at adaptation to OOD tasks in a drastically different domain such as P2A-OOD. Besides, similar to existing methods, PLP cannot handle novel groups of states that appear only in the target task, such as new objects. Moreover, to broaden the applicability of PLP, future work could explore approaches to utilize state-only demonstrations or add new domains to the existing representation space. Thanks to its simplicity, we can easily extend PLP to build our idea on top of it. We hope that our work provides valuable insights for researchers aiming to develop domain-free portable policies, abstract policies that can be applied to any domain in a zero-shot manner.

## REPRODUCIBILITY STATEMENT

We provide detailed descriptions of our experimental setups and implementations in Section 5.1 and Appendix C. These sections explain key hyperparameters, the design of environments and proxy tasks, dataset size, and other necessary information for reproduction. We also release our codebase and created datasets in the supplementary materials. The training time does not exceed about five hours with a single GPU in our experiment.

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

Table 3: Full table of the success rate comparison between ablated variants of PLP with standard deviations. The scores are averaged over nine seeds.

| Task | PLP-full | PLP-disc | BC + MMD | BC + disc |
|---|---|---|---|---|
| P2P-obs-medium | $0.92 \pm 0.09$ | $0.90 \pm 0.12$ | $\mathbf{0.94 \pm 0.05}$ | $0.91 \pm 0.08$ |
| P2P-medium | $0.84 \pm 0.08$ | $0.88 \pm 0.09$ | $\mathbf{0.93 \pm 0.05}$ | $0.81 \pm 0.13$ |
| P2A-medium | $\mathbf{0.70 \pm 0.17}$ | $0.52 \pm 0.15$ | $0.54 \pm 0.26$ | $0.45 \pm 0.21$ |
| R2R-Lift | $\mathbf{0.71 \pm 0.21}$ | $0.39 \pm 0.40$ | $0.63 \pm 0.37$ | $0.11 \pm 0.17$ |
| V2V-Reach | $0.68 \pm 0.24$ | $0.68 \pm 0.31$ | $\mathbf{0.70 \pm 0.07}$ | $0.37 \pm 0.40$ |
| V2V-Open | $\mathbf{0.68 \pm 0.14}$ | $0.33 \pm 0.11$ | $0.61 \pm 0.15$ | $0.00 \pm 0.01$ |

| Task | BC + TCC | BC Only |
|---|---|---|
| P2P-obs-medium | $0.87 \pm 0.10$ | $0.92 \pm 0.06$ |
| P2P-medium | $0.72 \pm 0.29$ | $0.48 \pm 0.23$ |
| P2A-medium | $0.62 \pm 0.24$ | $0.52 \pm 0.20$ |
| R2R-Lift | $0.54 \pm 0.42$ | $0.64 \pm 0.35$ |
| V2V-Reach | $0.43 \pm 0.33$ | $0.43 \pm 0.08$ |
| V2V-Open | $0.25 \pm 0.18$ | $0.00 \pm 0.01$ |

Table 4: The success rates of the Ant-to-Spider (A2S) task averaged over nine runs.

| Task | PLP (Ours) | GAMA | CDIL | Contextual | BC |
|---|---|---|---|---|---|
| A2S-umaze | $\mathbf{0.80 \pm 0.30}$ | $0.10 \pm 0.14$ | $0.00 \pm 0.00$ | $0.01 \pm 0.01$ | $0.42 \pm 0.38$ |
| A2S-medium | $\mathbf{0.45 \pm 0.18}$ | $0.04 \pm 0.05$ | $0.00 \pm 0.00$ | $0.00 \pm 0.00$ | $0.25 \pm 0.20$ |

# A  ADDITIONAL ILLUSTRATIONS, RESULTS AND DISCUSSIONS

## A.1  ALIGNMENT SCORE WITH TOP-5 ACCURACY

Figure 6 shows the alignment scores when we use top-5 accuracy instead of top-1 accuracy as a metric. The results show the same tendency as the one with top-1 accuracy (c.f. Figure 5).

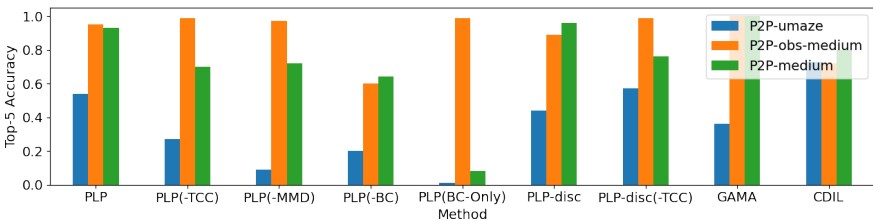

Figure 6: Alignment scores. The values are top-5 accuracies of finding a corresponding state from 1k random states based on the distance in the latent space (or state space for GAMA).

## A.2  FULL RESULTS OF ABLATION COMPARISONS OF LOSS TERMS.

Table 3 shows the full results with standard deviations.

## A.3  ADDITIONAL MAZE EXPERIMENTS WITH SPIDER ROBOT.

We test the transfer performance from an agent with four legs (Ant) to an agent with six legs (Spider). We call this task Ant-to-Spider (A2S). The robots are morphologically similar, but the position of the legs are different to each other. Table 4 shows the results. PLP-performs best among the methods.

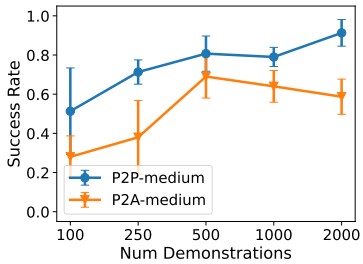 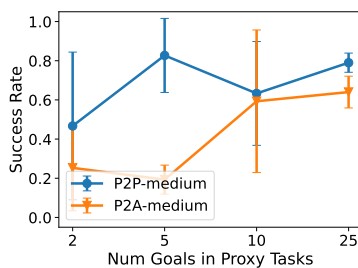

(a) The number of demonstrations vs success rate

(b) The number of goals in contained in proxy tasks vs success rate

Figure 7: Alignment complexity in the medium maze. The scores are averaged over three runs with a fixed single goal, and the error bars represent the standard deviations.

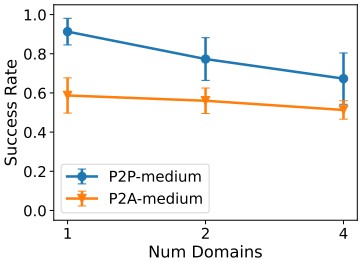 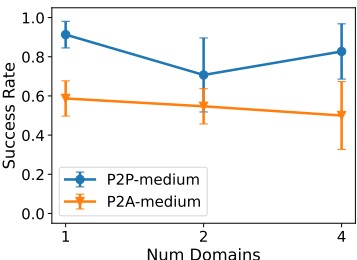

(a) The number of domains vs success rate, when the total number of trajectories is kept constant.

(b) The number of domains vs success rate, when the total number of trajectories increases proportionally.

Figure 8: The relationship between the number of source domains and the performance. The scores are averaged over three runs with a fixed single goal. The error bars show the standard deviations.

## A.4 ALIGNMENT COMPLEXITY

Figure 7 shows the alignment complexity in P2P-medium and P2A-medium with respect to the number of demonstrations and the number of proxy tasks. Note that the number of goals in the figure corresponds to one-fourth of the number of proxy tasks. We can observe that an increase in the number of demonstrations or proxy tasks positively affects performance, but the performance improvement does not continue until the perfect transfer especially in P2A seemingly due to the insufficient quality of alignment.

## A.5 EFFECT OF THE NUMBER OF SOURCE DOMAINS

One possible advantage of PLP compared to methods like GAMA is that we can easily increase the number of domains contained in the source dataset. However, it is unclear how the increase in domain variations affects imitation performance. To investigate this, we measure it in two settings: i) increase the number of domains in a fixed-size dataset ii) increase the number of domains while keeping the number of trajectories from one domain. In ii), the dataset size increases proportionally to the number of domains. For the comparison, we maintain the total number of epochs, as we observe that the training with the larger dataset does not converge within the same number of iterations.

The results measured in P2P-medium and P2A-medium are summarized in Figure 8. The increase in domain variations makes it difficult to shape a good shared representation space, resulting in a performance drop. As shown in Figure 9b, if the number of demonstrations for each domain is maintained, the performance degradation is mitigated, although it does not improve the performance compared to the single-domain case, either. Future work could explore methods that can effectively leverage multi-source datasets to boost the performance of a transferred policy.

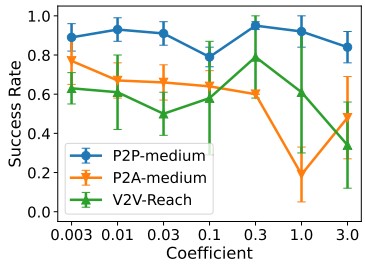
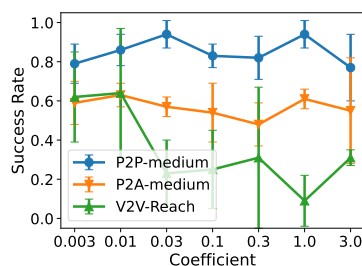

(a) The coefficient for MMD $\lambda_{\mathrm{MMD}}$ vs success rate.

(b) The coefficient for the discriminative loss in PLP-disc vs success rate.

Figure 9: The relationship between the coefficient of the regularization term and the performance. The scores are averaged over three runs with a fixed single goal. The error bars show the standard deviations.

Table 5: The success rates of PLP with the state input for the action decoder in R2R and V2V-Reach.

| Task | PLP | PLP + State | PLP + State (Proprio. only) |
|---|---|---|---|
| R2R-Lift | $\mathbf{0.71 \pm 0.21}$ | $0.53 \pm 0.48$ | N/A |
| V2V-Reach | $\mathbf{0.68 \pm 0.24}$ | $0.38 \pm 0.25$ | $0.64 \pm 0.30$ |

## A.6 EFFECT OF THE COEFFICIENT HYPERPARAMETER

Figure 9 shows the performance changes when we sweep the coefficient of the regularization term for the MMD of PLP or the discriminative objective of PLP-disc. We confirmed that the performance of PLP is not highly sensitive to the choice of $\lambda_{\mathrm{MMD}}$ For PLP-disc in V2V-Reach, we observe the performance degradation when we set the coefficient to the large value, so we use 0.01 for the experiments.

## A.7 PLP WITH STATE INPUT FOR DECODER IN R2R-LIFT AND V2V-REACH

As we mentioned in the last part of Section 4.1, the inclusion of state input for the decoder can sometimes lead to performance degradation. This occurs because the decoder can receive all necessary information except the task ID even without the common policy. While we show the performance without state input in the main results of R2R and V2V, here we present the performance with state input in Table 5. We observe a performance drop in environments, although it still significantly outperforms the baselines. For V2V-Reach, we also test a variant that exclusively receives proprioceptive inputs necessary for precise action prediction in the decoder, omitting the image inputs. This modification yields improved performance comparable to the full PLP. This highlights the potential for performance enhancement in PLP when we have prior knowledge about which inputs aid action prediction and which elements should be aligned across domains to facilitate better transfer.

## A.8 GAMA WITH STATE INPUT FOR DECODER

The reason why GAMA does not perform well in environments like P2A is the lack of symmetry and exact correspondence between domains, resulting in the failure of action translation of GAMA from the source domain to the target domain. In contrast, PLP overcomes this problem by learning a common latent space and focusing on transferring shared structure between domains. Another explanation of the performance gap between PLP and GAMA is that PLP provides domain-specific state information to the decoder while GAMA does not. To investigate the impact of state input to the performance of PLP, we measure the performance of GAMA when it receives state input for the action translation function of GAMA. The results are presented in Table 6. With the inclusion of the state input, GAMA is able to bridge the complexity gap between the source and target domain to some extent. However, it still faces difficulties in finding good correspondences between domains.

Table 6: The success rates of GAMA with the state input for the action translation function. Note that, in R2R-Lift, we do not provide states into the action decoder.

| Task | PLP | GAMA | GAMA + State |
|---|---|---|---|
| P2A-umaze | $\mathbf{0.50 \pm 0.41}$ | $0.01 \pm 0.03$ | $0.31 \pm 0.24$ |
| P2A-medium | $\mathbf{0.70 \pm 0.17}$ | $0.00 \pm 0.01$ | $0.11 \pm 0.07$ |
| R2R-Lift | $\mathbf{0.71 \pm 0.21}$ | $0.09 \pm 0.15$ | $0.21 \pm 0.31$ |

Table 7: Success rate with state-only source domain demonstrations

| Task | PLP | PLP-noaction |
|---|---|---|
| P2P-medium | $0.84 \pm 0.08$ | $0.17 \pm 0.20$ |
| P2A-medium | $0.70 \pm 0.17$ | $0.14 \pm 0.08$ |

### A.9 TRANSFER FROM STATE-ONLY SOURCE DOMAIN

We can extend the applicability of PLP to scenarios where actions from the source domains are unavailable. In such cases, we can replace action prediction with next state prediction. We decouple a PLP policy into next state prediction and inverse dynamics model. We calculate the loss on next state prediction in the representation space on the output of a common policy and optimize the encoder $\phi$ and common policy $\pi_z$.

$$\mathcal{L}_{\text{next\_state}} = \mathbb{E}\left[\|\pi_z(\phi_d(s_d), k) - \text{sg}[\phi_d(s'_d)]\|^2\right],$$

where $s'_d$ is a next state of $s_d$ in domain $d$, and $\text{sg}[\cdot]$ shows stopping gradient. At the same time, we separately train an inverse dynamics model $\phi^{-1}$ on the target domain dataset optimizing the following objective:

$$\mathcal{L}_{\text{IDM}} = \mathbb{E}\left[\|\psi^{-1}(\text{sg}[\phi_y(s_y)], s'_y) - a_y\|^2\right]$$

At inference, the encoder and common policy predict the next latent state given a current state, domain, and task ID, followed by the action prediction of the inverse dynamics model from a predicted next latent state and observed current state.

Table 7 presents the results. The performance is limited compared to the original setting. It is seemingly because this state-only setting misses the opportunity to leverage alignment signal from end-to-end behavioral cloning that the original PLP takes advantage of. Additional techniques are required to seamlessly bridge domains with and without action information. We leave it for future work as mentioned in the Conclusion.

## B ADDITIONAL VISUALIZATION

### B.1 DISTRIBUTION OF LATENT REPRESENTATION AND STATE CORRESPONDENCE

In Figure 10, we visualize corresponding states in the latent space of PLP obtained in R2R-Lift. With the help of MMD loss, the latent state distributions are overlapped. In addition, while there are instances of misalignment of states in some areas, we observe cross-domain state-to-state correspondence in arm positions.

In Figure 11, we visualized the latent state representation of PLP obtained in V2V-Reach. We color representation points for different ball orders with different colors in each domain how PLP organize and align the latent space. We observe that points for each color order form a cluster in the latent space, and the positions of clusters for the same arrangement of balls are located in a similar position across domains (11b, 11c).

### B.2 VISUALIZATIONS OF TRAJECTORIES OF AGENTS

We visualize the trajectories of the agents to observe their actual behavior. In Figure 12, we display the trajectories of PLP and Contextual in P2P-OOD. We confirm that PLP successfully follows the target route, while Contextual fails to adapt to this out-of-distribution target route. In Figure 13,

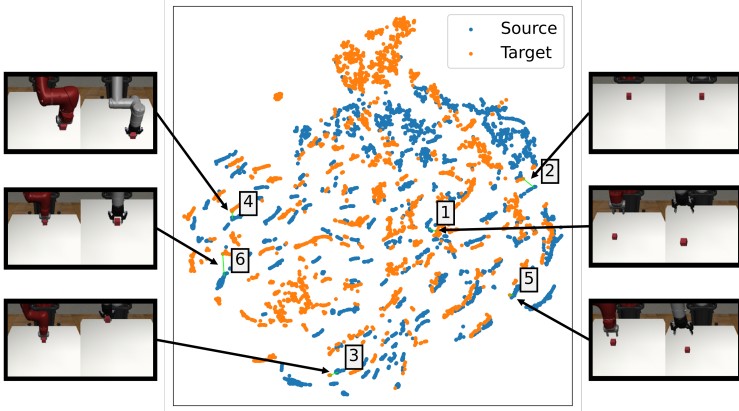

Figure 10: Visualization of the latent space by t-SNE in the R2R-Lift task. We visualize the states of the source (left-hand side) and target (right-hand side) domains that are closest to each other in the latent space. In some cases, the arms were in similar positions across domains (1, 2, 4, 5, 6), but in other cases, we observed positional deviation of the arms (3).

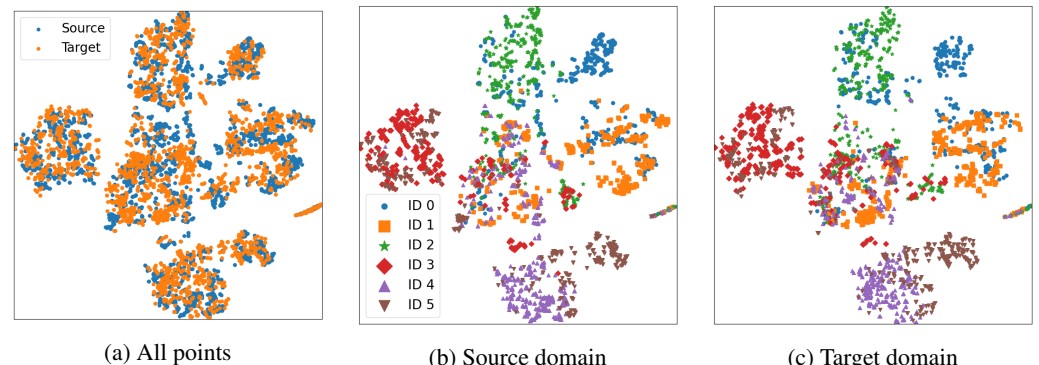

(a) All points          (b) Source domain          (c) Target domain

Figure 11: Distributions of latent states $s_z$ in V2V-Reach visualized by t-SNE. In (11b) and (11c), the points are colored according to the color order of the balls in the environment. ID 0-5 corresponds to GBR, GRB, BGR, BRG, RGB, and RBG, respectively. Each ID roughly forms a cluster in a consistent position across domains.

we present the trajectories of PLP and GAMA in P2A-medium. We observe that the PLP agent successfully reaches the goal, while the GAMA agent cannot move from the starting position due to the challenge of finding exact correspondence between the Point agent and the Ant agent.

Figure 14 demonstrates trajectories of PLP agent and GAMA agent in R2R-Lift. PLP agent precisely operates its arm and successfully grasps the target object, while GAMA agent attempts to close its gripper in the wrong position.

Figure 15 shows the trajectories of PLP agent and BC agent in V2V-Reach. PLP agent successfully adapts to the target task, while BC agent still heads to the goal of a proxy task. It is because the adaptation only happens on the source domain states as a BC agent learns about the source domain and target domain separately without aligning representations. We observe the same tendency in the trajectories of V2V-Open in Figure 16 as well. The PLP agent successfully recognizes the position of the window from the image and accomplishes the task (Figure 16a). The BC agent and Contextual agent stick to the movement in proxy tasks and move its arm in the opposite direction due to the lack of adaptation capability to OOD movements (16b, 16c).

We also provide videos of the agents in the supplementary material.

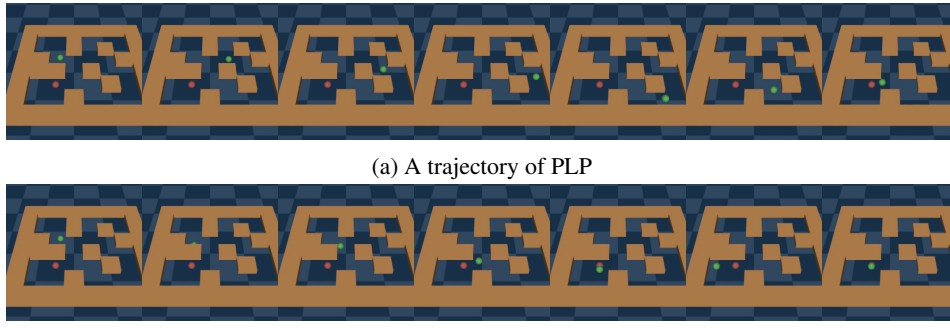

(a) A trajectory of PLP

(b) A trajectory of Contextual

Figure 12: Trajectories of agents in P2P-OOD. The red point shows the goal. The target route is illustrated by the red arrow in Figure 4f.

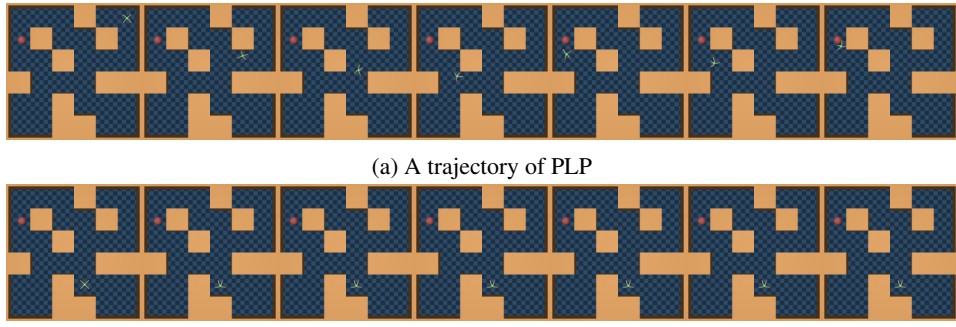

(a) A trajectory of PLP

(b) A trajectory of GAMA

Figure 13: Trajectories of agents in P2A-medium. The red point shows the goal.

## C  EXPERIMENT DETAILS

### C.1  ENVIRONMENTS

All Maze environments used in this paper are based on D4RL (Fu et al., 2020). These environments involve two types of agents: Point and Ant. The Point agent has a state space of four dimensions and an action space of two dimensions. The state space comprises the positions and velocities along the $x$-axis and $y$-axis, while the action space consists of the forces to be applied in each direction. The Ant agent has a state space of 29 dimensions and an action space of eight dimensions. The state space includes the position and velocity of the body, as well as the joint angles and angular velocities of the four legs. The action space consists of the forces applied to each joint. The task is defined by a combination of a starting area and a specific goal location. In the umaze, there are three starting zones and seven goals, while in the medium maze, there are four starting zones and 26 goals. For the experiments, three goals from different areas in the mazes are selected. In P2P, we constructed the source domain by swapping the $x$ and $y$-axis of observations and multiplying each element of actions by -1. For example, a state-action pair in the target domain $((x, y, v_x, v_y), (a_x, a_y))$ corresponds to $((y, x, v_y, v_x), (-a_x, -a_y))$ in the source domain. Here, $x$ and $y$ represent coordinate values instead of domains. In the OOD variants, agents are required to take a detour in the medium maze as depicted in Figure 4f, instead of directly heading toward the goal via the shortest path. In P2A, although the shapes of the mazes are consistent between Point and Ant, the scale and x, y directions of the mazes are different between agents from the beginning.

For R2R-Lift, we use robosuite framework (Zhu et al., 2020). The Sawyer robot and the UR5e robot are used for the source and target domain, respectively, to test cross-robot transfer (Figure 4c). The observation spaces of these robots consist of the positions or angular information, as well as the velocity of each joint and the end effector. Sawyer and UR5e have state spaces of 32 dimensions and 37 dimensions, respectively. Both robots are controlled by delta values of the 3D position of the

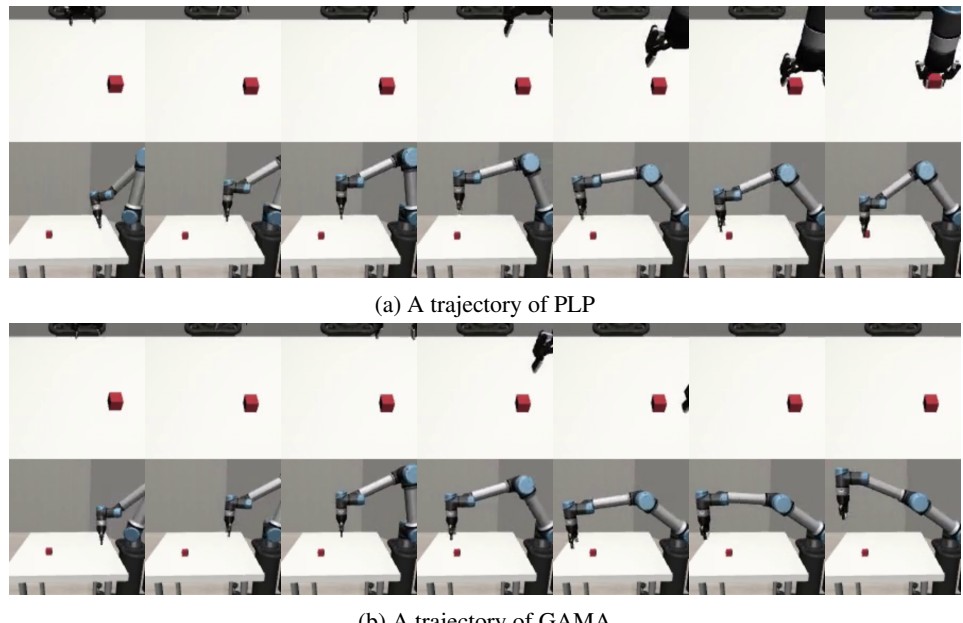

(a) A trajectory of PLP

(b) A trajectory of GAMA

Figure 14: Trajectories of agents in R2R-Lift.

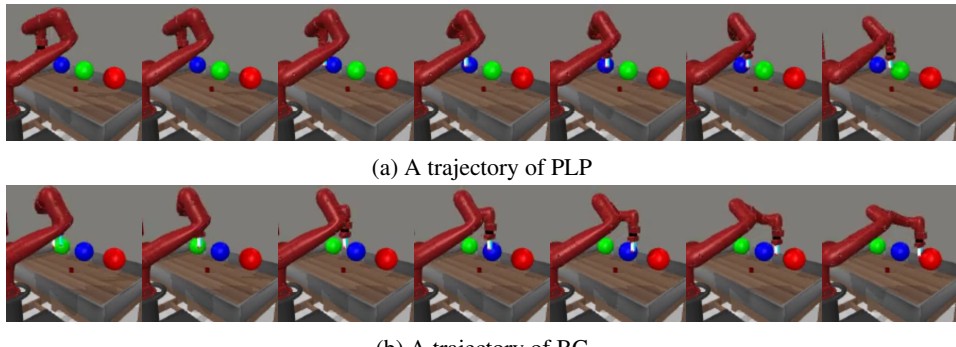

(a) A trajectory of PLP

(b) A trajectory of BC

Figure 15: Trajectories of agents in V2V-Reach. The goal color is set to green.

end effector and a 1D gripper state. We choose Block Lifting task, where the robot needs to pick up a block and lift it to a certain height. A task is defined by the initial position of the object to be lift. We set up a total of 27 tasks by placing the blocks on a $3 \times 3 \times 3$ grid of points on the table. One position is selected for the target task, and the remaining points that are not at the same height or the same 2D position as the selected position are used as the proxy tasks. The initial pose of the robot is randomized.

For V2V-Reach and V2V-Open, we use environments from the Meta-World benchmark (Yu et al., 2019). Different viewpoints are employed for the source and target domain as depicted in Figure 17. The robot observes the proprioceptive state and an image from a specific viewpoint. A proprioceptive state of the robot consists of a 3D position of the end effector and a 1D gripper state observed in the last two steps. The action space of the robot is 4-dimensional, representing the delta values of the 3D position of the end effector and the 1D gripper state. In V2V-Reach, three balls of different colors spawn in a random order. The task is to reach a ball of a specific color. The agent needs to interpret visual observation correctly. The initial position of the end-effector is also randomized.

In V2V-Open, we use Window-Close for the proxy task and Window-Open for the target task. Window-Close is the task of closing windows in the environment, while Window-Open is the task of opening windows. When closing the window, the robot approaches the left-hand side of a door

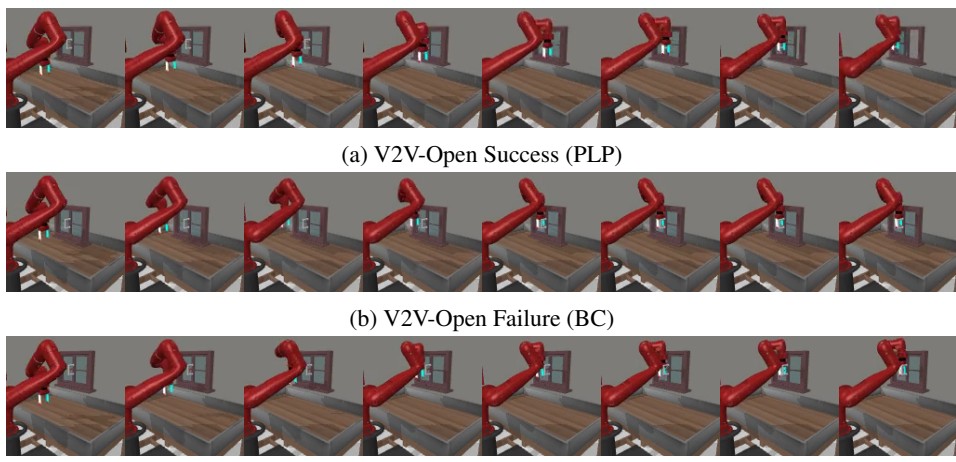

(a) V2V-Open Success (PLP)

(b) V2V-Open Failure (BC)

(c) V2V-Open Failure (Contextual)

Figure 16: Trajectories of agents in V2V-Open.

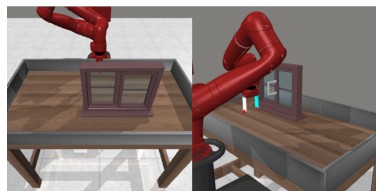

Figure 17: Visualization of two different viewpoints in V2V. The agent is required to learn a target task from one viewpoint and perform the task in another domain with a different viewpoint. The figure on the left shows the viewpoint in the source domain, while the figure on the right shows the one in the target domain.

and moves toward the right, whereas when opening the window, it heads to the right-hand side and then moves to the left. That is, the robot has to take out-of-distribution actions in the target task. Specifically, Window-Close is divided into four proxy tasks by roughly grouping the position of the window to facilitate representation alignment. The success rate of a proxy task (i.e., one area) without visual input is approximately 30%. In V2V-Open, which is a target task, we sample the window position from the entire area randomly.

In the evaluation, we measure the success rates with 100 trials in each seed, and we show the average and standard deviations calculated over nine seeds.

## C.2 DATASETS

As explained in Section 3, the dataset comprises state-action trajectories of expert demonstrations encompassing multiple tasks with different goals. Specifically, for the P2P and P2A, we provided about 1k trajectories from 18 tasks (six training goals and three starting zones) for the umaze, and about 1k trajectories from 100 tasks (25 training goals and four starting zones) for the medium maze unless stated otherwise in the ablation study. The expert demonstrations of the Point agent were downloaded from http://rail.eecs.berkeley.edu/datasets/offline_rl/maze2d/ (maze2d-umaze-sparse-v1, maze2d-medium-sparse-v1). When generating expert trajectories for the Ant agent, we employed PPO (Schulman et al., 2017) from stable-baselines3 (Raffin et al., 2021) to train agents to move one of the four cardinal directions (up, down, left, or right). Subsequently, we constructed complete demonstrations by solving the maze using breadth-first search (BFS) and providing the agent with the direction of the next square. For R2R and V2V tasks, we collected expert trajectories with scripted policies based on the object position and the gripper pose. For R2R, we provide 1k trajectories in total in each domain. For V2V-Reach, we provided 300 trajectories for each task (i.e., each goal color). For V2V-Open, we provided 300 trajectories for

Window-Close in total, that is, 75 trajectories for each. In adaptation, we provide 300 trajectories of Window-Open.

### C.3 ARCHITECTURES AND TRANING DETAILS OF PLP

Our policy architecture is a simple multilayer perceptron (MLP). The state encoder, common policy, and decoder consist of three, four, and three hidden layers, respectively, with 192 units each and GELU activations (Hendrycks & Gimpel, 2016). The final layer of the decoder uses Tanh activation. The dimension of the latent representations is also set to 192. For the visual inputs in V2V, we used a convolutional network with CoordConv (Liu et al., 2018a) of channels (64, 64, 128, 128, 128), kernel size $= 3$, stride $= 2$, and padding $= 1$, followed by a single linear layer that projects the output into a single vector with 1024 dimensions. In order to keep visual information that is irrelevant for proxy tasks but necessary for the target task, we add an image reconstruction loss to the objective.

We optimized our objective with AdamW optimizer (Loshchilov & Hutter, 2019). In the Maze environments, we set the learning rate to 5e-4 and the batch size to 256, and trained the model for 20 epochs. In the adaptation phase, we used about 400 trajectories towards an unseen goal to update the common policy for 50 epochs. For R2R-Lift, we set the learning rate to 1e-4 and the batch size to 512, and trained the model for 100 epochs. For V2V-Reach, we set the learning rate to 5e-4 and the batch size to 64, and trained the model for 100 epochs. For V2V-Open, we set the learning rate to 1e-4 and the batch size to 64, and trained the model for 200 epochs. As mentioned in Section 4.2, we mixed the dataset for the alignment so that domain-specific component does not leak into the common policy. We randomly selected data from the alignment dataset of twice the size of the adaptation dataset in P2P and R2R, and that of the same size for P2A and V2V. We also found that adding weight decay stabilizes the alignment and the performance of P2P and P2A. We set the coefficient to 1e-5 in these environments. We did not use weight decay in R2R and V2V. For a kernel for MMD, we use Gaussian kernel: $f(a,b) = \exp(-\|a-b\|^2/h)$. $h$ is set to 1 in all experiments. Since we observed that it decreased the scale of representation $s_z$ and deteriorated the performance, we normalized the distance $\|a - b\|$ by the average pairwise distance in a batch to mitigate the issue. For the TCC part, we reduced the batch size to 64, 32, and 8 in the Maze, R2R, and V2V environments respectively to reduce training time. The number of gradient steps for the TCC part is kept the same as the other parts. Additionally, to alleviate the difficulty of classification in TCC, we decimated the trajectories by selecting one out of every 16 frames. Regarding the coefficients in the objective 3, we set $\lambda_{\text{MMD}} = 0.1$, $\lambda_{\text{TCC}} = 0.2$ for P2P, P2A, and R2R, and $\lambda_{\text{MMD}} = 0.01$, $\lambda_{\text{TCC}} = 0.1$ for V2V environments. In PLP-disc for the ablation, where we utilized the GAN-like discriminative loss instead of MMD regularization, we introduced a discriminator network with four hidden layers. The coefficient for the discriminative loss was set to 0.5 for P2P, P2A, and R2R, while we used 0.1 for V2V experiments. The training time was about an hour for the maze environments and R2R, four hours for R2R-Lift, and three hours for the V2V environments with a single GPU.

In the supplementary material, we provide our implementation for reproduction.

### C.4 BASELINES

For GAMA, we re-implemented the algorithm by referring to the original paper and an official implementation (`https://github.com/ermongroup/dail`). When we found discrepancies between the paper and the implementation, we followed the descriptions in the paper. We swept the adversarial coefficient from 0.01 to 10, the learning rate from 1e-4 to 1e-3, and selected 0.5 and 1e-4, respectively.

For CDIL, we re-implemented the algorithm based on the paper. First, we pretrained the temporal position models in both the source and target domains. Subsequently, we trained all models using cycle consistency loss, adversarial loss, and temporal position loss between domains. At the same time, we performed inference task adaptation using data from the target task in the source domain. Next, we trained the inverse dynamics model in the target domain. Once the target task data in the source domain was converted into the target domain, the inverse dynamics model was used to compute the target task actions in the target domain. Finally, the final policy was obtained through behavioral cloning using this data.

For the demonstration-conditioned model (Contextual), we used a Transformer (Vaswani et al., 2017)-based architecture to process sequences of observations and actions. We fed the demonstrations and observation history as they were without thining-out timesteps. The maximum sequence length was 250 and 400 for the umaze and medium maze in P2P, 350 and 600 in P2A, and 200 in R2R and V2V, respectively. The model had three encoder layers and three decoder layers, each with 256 units and eight heads. The dropout rate was set to 0.2. The activation function was GELU and the normalization was applied before the residual connections. We set the batch size to 64, the learning rate to 1e-3, and trained the model for 500 epochs in each environment in Maze environments and R2R. In V2V, we set the batch size to 16.

