# OpenReview forum: "Leveraging Behavioral Cloning for Representation Alignment in Cross-Domain Policy Transfer"
_ICLR.cc/2024/Conference — Submitted to ICLR 2024_

### Official Review · Reviewer_QnVM · 2023-10-29

**Soundness:** 2 fair
**Presentation:** 3 good
**Contribution:** 2 fair
**Rating:** 5
**Confidence:** 4

**Summary:**

This paper presents a new method named Portable Latent Policy (PLP) for cross-domain policy transfer and demonstrates the effectiveness of the proposed method on standard robot control benchmarks such as D4RL, in both cross-robot and cross-viewpoint scenarios. The technical contribution of PLP is to leverage multi-domain behavioral cloning to align latent state representations across different domains. The key idea is to enable the transfer of policies trained on the shared representation.

**Strengths:**

1. Quality: This paper is well-structured and provides a clear and detailed description of the proposed method and the results. Despite the diverse experimental settings, most contents are clear and easy to follow.
2. Novelty: Overall, the core idea of multi-domain behavioral cloning via latent state alignment is novel for cross-domain policy transfer (although individual components like MMD and TCC are not new). Further, it is practical, as it avoids the need for labeling the exact correspondence of the cross-domain trajectories.
3. Experimental findings: The paper provides solid empirical findings by evaluating the proposed method in various domain transfer scenarios. The results show that PLP outperforms the existing methods.
4. Good reproducibility.

**Weaknesses:**

1. In Table 1, the proposed model commonly has a larger standard deviation than the compared methods. It would be good if the authors could provide more analysis of these results. Besides, I wonder if PLP (including PLP-disc) is sensitive to the hyperparameter choices.
2. In Table 2, the ablated variant of the proposed model, BC + MMD, outperforms PLP-full on three out of six tasks. This raises a question about the necessity of the TCC component in the overall framework. Further clarification on the role and impact of TCC in improving performance across various tasks would provide valuable insights.
3. The method's effectiveness appears to be closely tied to the similarity between the source and target domains. However, the paper lacks an in-depth analysis of this issue concerning PLP's performance in various practical scenarios. It would be valuable to explore how the method performs when dealing with expert demonstrations solely from a highly dissimilar source domain, as well as when utilizing demonstrations exclusively from the target domain (where the source and target domains are identical). Additionally, the paper could benefit from further discussions regarding strategies for identifying a reliable source domain.
4. A minor issue: The description in the caption for Figure 4(e) appears to correspond to Figure 4(f).

**Questions:**

My major concern with this paper is about the provided experimental results. Please see the comments above.

---

> ### Author Response · Authors · 2023-11-23
>
> Thank you for reviewing our paper.
>
> > In Table 1, the proposed model commonly has a larger standard deviation than the compared methods. It would be good if the authors could provide more analysis of these results. Besides, I wonder if PLP (including PLP-disc) is sensitive to the hyperparameter choices.
>
> The score depends highly on the success of feature alignment or correspondence learning for domain translation. The scores are not distributed around the mean values but take either low or high values. So PLP or other methods have a larger distribution when successful (i.e., a higher one takes a closer value to 1.0).
> Besides, we added an ablation study for hyperparameter choices to Appendix A.6 and Figure 9.
> We observe that the performance was not highly sensitive to the MMD coefficient.
>
> > In Table 2, the ablated variant of the proposed model, BC + MMD, outperforms PLP-full on three out of six tasks. This raises a question about the necessity of the TCC component in the overall framework. Further clarification on the role and impact of TCC in improving performance across various tasks would provide valuable insights.
>
> It is natural that the information on task progression brought by TCC is task-dependent.
> TCC might not necessarily be included in PLP-full for all tasks, but we keep it in the method considering its contribution to alignment observed in Figure 3e and Figure 5 for this submission.
>
> > The method's effectiveness appears to be closely tied to the similarity between the source and target domains.
>
> This is true. As long as we try to perform offline cross-domain transfer without further exploration, the source domain and target domain have to be similar to some extent.
>
> > It would be valuable to explore how the method performs when dealing with expert demonstrations solely from a highly dissimilar source domain,
>
> In this paper, we focused on the simplest situations where existing methods even struggle, but it would be interesting to examine the applicability of the method in more advanced settings. As long as the structures of tasks are similar and the proxy tasks cover necessary states, the method should work given a sufficient amount of data.
>
> > as well as when utilizing demonstrations exclusively from the target domain (where the source and target domains are identical)
>
> We do not get your point about this. Isn’t it a simple imitation or demonstration-guided learning within a single domain?
>
> > A minor issue
>
> Thank you for pointing it out. We fixed it.

---

### Official Review · Reviewer_9N3r · 2023-11-01

**Soundness:** 2 fair
**Presentation:** 2 fair
**Contribution:** 2 fair
**Rating:** 5
**Confidence:** 4

**Summary:**

The paper proposes a method (PLP) for aligning latent state representations across different domains using unaligned trajectories of proxy tasks. It leverages multi-domain behavioral cloning to shape a shared latent space, as well as a few regularization terms (i.e., MMD and TCC) to further encourage cross-domain alignment, allowing policies trained on the shared representation to be transferred to another domain without the need for exact cross-domain correspondence or translation. The method is found to outperform existing methods that employ adversarial domain translation in various domain shift scenarios, including cross-robot and cross-viewpoint settings.

**Strengths:**

- PLP does not require discovering the exact cross-domain correspondence as a common practice well adopted in prior works, making it more flexible and applicable in various scenarios.
- PLP reflects on the widely-held adoption of domain classifier for distribution matching, demonstrating MMD helps with aligning latent distribution within a more modest manner and largely keeps the original representation structure.
- The method outperforms existing baselines that employ adversarial domain translation in different domain shift scenarios by a fairly observable margin.

**Weaknesses:**

- Novelty: learning a canonical representation space, as well as MMD and TTC, has already been applied in many transfer learning and cross-domain settings, not to mention that TCC actually proves to trivially contribute to performance improvement in experiement section. In the design of PLP, I simply see a combination of previous well-founded thoughts, however, expecting to see something more distinctively insightful to shed light on the avenue of cross-domain policy transfer where over recent years people have failed to deliver something new from a technical angle.
- Formula misalignment: the formulation of $\mathcal{L}\_{TCC}$ simply involves two encoded trajectories, which poses a mismatch with $\mathcal{L}\_{BC}$ and $\mathcal{L}\_{MMD}$ that are in expectation format over datasets.
- Shorthand misclaim: authors explains P2P and P2A in Figure 3 neither in the caption nor in the paragraph they first mention these shorthanded parlances in the main text (P5), but eventually in the experiment section, which would probably make readers confused at first.
- Seemingly symbol misclaim: inner optimization variable "q" appears to be missed to claim about what it means.
- Citation misspecification: "Kevin Zakka, Andy Zeng, Pete Florence, Jonathan Tompson, Jeannette Bohg, and Debidatta Dwibedi. XIRL: Cross-embodiment inverse reinforcement learning. In Conference on Robot
Learning, pp. 537–546. PMLR, 2022." should be CoRL 2021 acceptance instead of 2022.

**Questions:**

Please refer to weakness for details

---

> ### Author Response · Authors · 2023-11-23
>
> Thank you for reviewing our paper.
>
> >  expecting to see something more distinctively insightful to shed light on the avenue of cross-domain policy transfer where over recent years people have failed to deliver something new from a technical angle.
>
> We agree with your point. We conjecture the relatively slow progress in this area can be partially attributed to the complicated structure of methods which makes them harder to extend. This paper aims to organize and simplify the necessary components to tackle various domain shifts by demonstrating that our minimalistic approach actually outperforms previous approaches, expecting future extensions added to the simple framework.
>
> > Formula misalignment
>
> > misclaim
>
> > misspecification
>
> Thank you for pointing the issues out. We fixed them.

---

### Official Review · Reviewer_EC2z · 2023-11-04

**Soundness:** 3 good
**Presentation:** 2 fair
**Contribution:** 2 fair
**Rating:** 3
**Confidence:** 3

**Summary:**

The paper presents a novel approach to cross-domain transfer learning, specifically within the context of reinforcement learning and imitation learning. The core innovation of the paper is a method that enables knowledge transfer across different domains without the need for exact one-to-one correspondence between those domains. This is particularly useful in scenarios where the domains are significantly different in terms of observation space, action space, or agent morphology.

To achieve this, the authors propose a technique that leverages a shared latent representation space. This space is crafted using signals from multi-domain imitation learning and is further refined with domain confusion regularization via Maximum Mean Discrepancy (MMD). This contrasts with previous methods that heavily relied on domain translation techniques. It requires only a policy network and a few loss terms for optimization, making it more straightforward than other offline cross-domain transfer methods that typically need more models and objectives.

The authors evaluate their approach across a variety of domain shifts, including changes in observation, action, viewpoint, and agent morphology. The results demonstrate that their method outperforms existing methods, especially in situations where an exact translation of states or actions is difficult or impossible to achieve. The paper also shows that the method has superior adaptation capabilities for out-of-distribution tasks and that MMD regularization performs better than discriminative training approaches in these contexts.

**Strengths:**

1) The authors innovatively address the challenge of cross-domain policy transfer by adapting unsupervised domain adaptation techniques to the reinforcement learning paradigm, showcasing a novel intersection of methodologies.
2) The paper presents a comprehensive evaluation of the proposed methods across complex transfer scenarios, including cross-robot and cross-viewpoint transfers, demonstrating the method's versatility and effectiveness in diverse environments, including the adaptation capabilities for out-of-distribution tasks.
3) The proposed method leverages a shared latent representation space, which facilitates the transfer of knowledge without the need for direct domain correspondence, thus overcoming a significant limitation in traditional domain adaptation, such as more robust hyperparameters and training instability, reducing computational overhead and simplifying the implementation.

**Weaknesses:**

My concerns focus on the possible novelty issues and motivations.
1) The paper posits that existing domain translation methods struggle with large discrepancies between domains and may not provide stable cross-domain alignment. However, this claim lacks empirical evidence or a theoretical foundation within the paper, which weakens the argument. A comparative analysis or quantitative evidence would strengthen the case for the proposed method's necessity and superiority. Besides, in general, we conduct the domain adaptation when there is no huge distribution shift.  For example, in [1], the target shift is difficult to solve if there are no further assumptions. The examples shown in the introduction parts have less practical application significance, such as the adaption from the agent with no legs to other agents with multiple legs.
2) For this toy motivational example, despite acknowledging this issue, this paper doesn't involve it in their experimental evaluations. Including such a scenario, especially one that has been previously explored in the literature, would not only validate the method's effectiveness in extreme cases but also provide a direct comparison to existing solutions. This toy experiment has been explored in [2].
3) The paper could benefit from a more thorough discussion of other methods that tackle generalization in reinforcement learning. By comparing the proposed method to these existing approaches, the authors would better situate their work within the current research landscape and potentially highlight the unique advantages of their approach. For example, [2,3,4,5] also explores the generalization problem in the reinforcement learning field.
4) The potential novelty issue.  This method is based on MMD, which is widely used in domain adaptation. What is the unique contribution to introducing MMD into reinforcement learning? Do the weaknesses in MMD still exist in this field, such as MMD's sensitivity to kernel choice and its performance in high-dimensional spaces?
5) While the method claims to be computationally simpler, the paper does not provide a detailed analysis of computational costs or training times compared to other methods. Such an analysis would be beneficial for practical applications where computational resources and efficiency are critical factors.



[1]Zhang K, Schölkopf B, Muandet K, et al. Domain adaptation under target and conditional shift[C]//International conference on machine learning. PMLR, 2013: 819-827.
[2]Zhang A, McAllister R, Calandra R, et al. Learning invariant representations for reinforcement learning without reconstruction[J]. arXiv preprint arXiv:2006.10742, 2020.
[3]Tomar M, Zhang A, Calandra R, et al. Model-invariant state abstractions for model-based reinforcement learning[J]. arXiv preprint arXiv:2102.09850, 2021.
[4]Mahajan A, Zhang A. Generalization Across Observation Shifts in Reinforcement Learning[J]. arXiv preprint arXiv:2306.04595, 2023.
[5]Szot A, Zhang A, Batra D, et al. BC-IRL: Learning Generalizable Reward Functions from Demonstrations[J]. arXiv preprint arXiv:2303.16194, 2023.

**Questions:**

As a reviewer with expertise in domain adaptation rather than reinforcement learning, my evaluation of this paper is primarily grounded in the principles and challenges specific to domain adaptation. My current assessment may evolve following a detailed discussion with the authors and other reviewers to ensure a comprehensive understanding of the paper's contributions.

The critical points of my review, detailed in the Weaknesses Section, revolve around the foundational motivation for the proposed method and questions regarding its novelty and empirical support. Specifically: 1) Why do existing domain translation methods struggle with large discrepancies? 2) How can we verify this? 3) Can this method be used for the toy example shown in the introduction?  4) What is the unique contribution beyond MMD and what are the advantages of this method over [2,3,4,5]?

---

> ### Author Response · Authors · 2023-11-23
>
> Thank you for reviewing our paper.
>
> > The paper posits that existing domain translation methods struggle with large discrepancies between domains and may not provide stable cross-domain alignment. However, this claim lacks empirical evidence or a theoretical foundation within the paper, which weakens the argument.
>
> We extensively conduct comparison experiments between PLP and domain-translation-based methods (GAMA and CDIL), although we do not provide any theoretical support for it.
>
>
> > For this toy motivational example, despite acknowledging this issue, this paper doesn't involve it in their experimental evaluations.
>
> In the example, we intended to represent the Point2Ant example, so we have comparison results in the experiment section. As for the experiments in [2], it’s not trivial to use it in our experiments since we need multiple tasks to construct proxy task datasets, though we find it useful for constructing multiple domains easily.
>
> > What is the unique contribution to introducing MMD into reinforcement learning?
>
> Unfortunately, we don’t have a clear answer to it. The community somehow preferred GAN-like domain-discriminative training over the years. This paper demonstrates that, at least in our use case of representation regularization on domain-shared latent space rather than on raw states and actions, MMD can help better in shaping a latent structure especially when the domains are not symmetric.
>
> >  the paper does not provide a detailed analysis of computational costs or training times
>
> We do not claim the computational efficiency of PLP. We claim PLP is much simpler than existing domain-translation methods in terms of the number of loss terms and accompanying coefficients and the number of necessary models. The simplicity of PLP makes it amenable to possible future extensions.
>
> > 1) Why do existing domain translation methods struggle with large discrepancies?
>
> Without supervision for correspondence, it can be difficult to find cross-domain correspondence via adversarial domain translation. It might be different in situations like style transfer. In this sense, it’s reasonable for GAMA and CDIL to work well in P2P. We have no good explanation for why these methods did not work well in R2R.
>
> > 2) How can we verify this
>
> GAMA and CDIL are based on domain translation, and we empirically show PLP’s superiority over these methods. In particular, GAMA and PLP-disc are architecturally close to each other and the main difference lies in the difference between domain translation vs representation alignment using multi-domain BC.
>
> >  4) What is the unique contribution beyond MMD and what are the advantages of this method over [2,3,4,5]?
>
> First of all, the introduction of MMD is not the main contribution of our paper. The main point is the utilization of signals from multi-domain BC, which results in our simple architecture.
> As for the comparison to DBC-based methods [2,4] and state abstraction with dynamics modeling [3], one obvious difference seems to be that these methods rely on the identicality of action spaces.
> PLP only utilizes expert demonstrations that should correspond to high-reward actions, whereas DBC-based ones need dense reward signals for every state and every task to find an important structure of state space. DBC also models transition dynamics and PLP might benefit from it as well. For [5], we do not find high relevance to our approach at the moment.

---

### Official Review · Reviewer_WwQw · 2023-11-09

**Soundness:** 3 good
**Presentation:** 3 good
**Contribution:** 3 good
**Rating:** 5
**Confidence:** 3

**Summary:**

The paper addresses the problem of transferring learned policies across domains. To this end, the paper proposes to align latent state representations across different domains using unaligned trajectories of proxy tasks and an MMD based loss. The proposed method is evaluated across various domain shifts, including cross-robot and cross-viewpoint settings.

**Strengths:**

• The proposed PLP approach is novel and interesting. The core idea of having a common latent space aligned using MMD is promising.

• The paper is well written. Figure 1 provides a good overview of the proposed approach.

• The proposed two phase alignment and adaptation training scheme is novel.

• Results in Figure 3 show clearly that the proposed model training with MMD is able to learn a common latent space.

• Results in Table 1 show that the proposed PLP approach is able to successfully perform cross domain policy transfer in complex environments.

**Weaknesses:**

• Relation to work in generative modelling: The proposed cross modal domain alignment architecture is very similar to “Latent Normalizing Flows for Many-to-Many Cross-Domain Mappings, ICLR 2020”. This work also includes domain specific encoders/decoders and a shared common latent space aligned using a INN.

• The paper should discuss the evaluation datasets/tasks in more detail. E.g. for the R2R-Lift task, the paper should explain in more detail the action spaces of the source and target domains. The paper only mentions that “a robot has to learn from another robot with a different number of joints and different grippers”. How much of a change in the action space / number of grippers can the proposed PLP method deal with?

• In Figure 5, the paper should explain in more detail the factors that effect the alignment across tasks. E.g., why is there is a significant difference in the alignment between P2P-umaze and P2P-medium?

• Application to real-world tasks such as autonomous driving: The paper focuses mostly on synthetic tasks. It would be interesting to see if the proposed approach is applicable to real-world tasks such as autonomous driving e.g. transferring from CARLA to nuScenes.

**Questions:**

• A more detailed discussion of related work will be very helpful.
• The paper should also discuss the evaluation datasets/tasks in more detail.

---

> ### Author Response · Authors · 2023-11-23
>
> Thank you for reviewing our paper.
>
> > Relation to work in generative modelling
>
> Thank you for sharing the interesting related paper. The architecture they proposed seems somewhat similar to ours, but in our case, we cannot utilize supervision of one-to-one correspondence between modalities, which makes the problem more difficult.
>
> > The paper should discuss the evaluation datasets/tasks in more detail. E.g. for the R2R-Lift
>
> The details are described in Appendix C.1. For example, Sawyer and UR5e robots have a 32-dim state and 37-dim state, respectively with the same format of action space, although the outcome of action, especially that of “close gripper”, is similar but not identical because the shape of gripper differs (Figure 4c).
>
> >  the factors that effect the alignment across tasks.
>
> It is actually not completely clear to us. On the difference in the alignment score between P2P-umaze and P2P-medium, the smaller number of proxy tasks might make alignment a little inaccurate. As for the downstream performance, we observe that it failed only when the target task was at the end of the maze. The agent successfully moved to the place right before the goal and failed to proceed to the goal. It is probably because the dataset for the alignment only contains trajectories of an agent moving from the goal place to the center as we remove the demonstrations for the target task in the alignment phase. The method might still not be very robust to out-of-distribution actions, although it can successfully handle such actions in V2V-Open.

---

### Official Review · Reviewer_NwUM · 2023-11-09

**Soundness:** 2 fair
**Presentation:** 2 fair
**Contribution:** 2 fair
**Rating:** 3
**Confidence:** 3

**Summary:**

The paper presents a novel approach to align representations between two robotic domains by training an encoder, a policy based on an MDP, and a decoder. The method is tested in downstream tasks, showing promise in its primary objectives. However, it requires additional experimentation to substantiate its claims and illustrate its significance fully.

----- Edit -----

I have read the author's response. I appreciate they helped clarify my questions. Unfortunately, the provided empirical evidence fail to illustrate the strength of their contributions. The concern I have about the existing evaluations is that: the choice of Point2Ant seems arbitrary, and need more systematic evaluation to show its generalization (Point2Hopper, Ant2Hopper, Hopper2Ant, Point2Walker, Ant2Walker, etc). To the authors' credits, they added Ant2Spider, which is an interesting evaluation. Nevertheless, I think adding more evaluation across more tasks would be essential to prove the proposal's generalization. So I chose to keep my score.

**Strengths:**

- Demonstrates an improvement over existing baselines on some tasks.
- Tackles a convincing problem statement that is relevant and well-articulated.

**Weaknesses:**

The test environments selected are overly simplistic and appear arbitrary, which undermines the potential impact and applicability of the proposed methods.

**Questions:**

1. Clarify the environment setup. Specifically, define the observational shift in P2P-obs and detail the P2A action space for both Point and Ant agents.
2. Expand the variety of domain transfers. This paper completed P2P and P2A. But we have never had a robot resembling a Point robot in the real world. Why not Ant2Hopper? Can you run more combinatorial options: Ant2Hopper, Ant2Walker2D? Can you include multiple source domains e.g., Ant, Hopper, Walker2D and HalfCheetah? With the promise of a general alignment method, it is necessary see how well the method perform on general and more realistic domains. While P2P and P2A transfers have been completed, the absence of a Point robot raises questions. Exploring transfers like Ant2Hopper or Ant2Walker2D and including multiple source domains (e.g., Ant, Hopper, Walker2D, HalfCheetah) would significantly showcase the generalizability of the method across more realistic domains.
3. Reevaluate the manipulation task in the R2R-Lift scenario. The current evaluation R2R-Lift is completed on end effector delta space. So the task setup combines (1) tabletop manipulation task like lift where the robot barely needs to rotate the wrist, (2) in end effector delta space, (3) using low dimensional vector space representation. The current setup is too simplistic to conclusively determine the method's effectiveness. Introducing tasks that require wrist rotation, controlling manipulators over joint space, or running R2R in the visual space could provide a more robust assessment.
4. V2V uses the same robot in the transfer. In this case, the encoder is essentially learning a viewpoint transformation. Address the V2V task's potential overlap with existing viewpoint transformation methods. A comparison with state-of-the-art viewpoint alignment methods could highlight the unique contributions of your work.
5. The proposal is learning two modules, encoder and decoder. It could help increase the impact of this paper to design evaluations to separately test how well your proposal is learning the encoder (e.g. your V2V task where the visual inputs are from different perspectives) and decoder (e.g. transfer between two domains with non-trivial action space).
6. Increase the robustness of statistical claims by conducting more trials. Ideally, such simulator task should be evaluated with ≥3 different random seeds across ≥20 runs.
7. Discuss the assumption that multiple MDPs share a common latent structure. The paper should acknowledge the potential risks of aligning distributions that do not naturally overlap, and if possible, provide an argument against this critique.
8. Explore and discuss alternative loss functions. The motivation behind the choice of losses is clear, yet an exploration of alternatives could be insightful for the reader and the field.

---

> ### Author Response · Authors · 2023-11-23
>
> Thank you for your review and helpful suggestions.
>
> > Clarify the environment setup. Specifically, define the observational shift in P2P-obs and detail the P2A action space for both Point and Ant agents.
>
> As we mentioned in the main script, the details of environments are available in Appendix C.1.
> In P2P-obs x and y-axis of observation are swapped. In P2A, Point outputs 2-dim actions for x and y axes, whereas the Ant agent is controlled by 8-dim actions for joints.
>
> > Why not Ant2Hopper? Can you run more combinatorial options: Ant2Hopper, Ant2Walker2D?
>
> In these standard benchmarks such as HalfCheetah and Hopper, it’s not easy to create multiple distinct tasks as the agents are only moving forward if we understand correctly. One option would be using move forward and backward as a proxy task and target task, respectively.
>
> > Exploring transfers like Ant2Hopper or Ant2Walker2D … would significantly showcase the generalizability of the method across more realistic domains.
>
> We added the results of Ant2Spider to the Appendix, although the morphologies are not largely different.
>
> > V2V uses the same robot in the transfer.
>
> As long as the action space and dynamics are not largely different, it is not difficult to use different robots for the experiments, as the appearance of the scene already has a significant difference between domains.
>
> > Address the V2V task's potential overlap with existing viewpoint transformation methods
>
> In the context of cross-domain transfer/adaptation, cycle consistency is one main approach to the viewpoint shift. We choose CDIL from such a method as it claims it outperforms simple cycle consistency-based domain translation.
> We don’t argue that our method outperforms methods that are specifically designed for viewpoint transfer with the supervision of viewpoint correspondence in a scene. It is posed as one example of domain shift in observation space.
>
> > Reevaluate the manipulation task in the R2R-Lift scenario
>
> Although P2A (and Ant2Spider added to the Appendix) includes control in a more complex action space, it would be valuable to confirm it in manipulation tasks as well.
> Control based on the pixel observation is covered by V2V experiments.
>
> > Increase the robustness of statistical claims by conducting more trials. Ideally, such simulator task should be evaluated with ≥3 different random seeds across ≥20 runs.
>
> We apologize for not clearly stating it. We mentioned the scores are averaged over nine runs, but in each seed, we calculated an average of 100 runs. The large standard deviations are not due to the small number of runs but the outcomes of genuine unstableness, unfortunately. The error of the mean estimation is much lower than the values shown in the tables. We added this to the paper.
>
> >  The paper should acknowledge the potential risks of aligning distributions that do not naturally overlap
>
> We think we cannot completely remove this risk as the other methods cannot. As we increase the coefficient of the MMD term, the method would enforce the overlap ignoring its effect on action prediction (i.e., BC term). We can mitigate the issue by encouraging more state-to-state alignment either in a supervised or unsupervised way such as TCC.
>
> >  an exploration of alternatives could be insightful for the reader and the field.
>
> We have explored a few more novel alternatives, but only the discriminative loss and the MMD worked consistently well, even in P2P/P2A.

---

### Meta-Review · Area_Chair_D9k5 · 2023-12-05

**Metareview:**

(a) Summary
The goal of the paper is to learn a robot policy that can be transferred across domains. To do so, they learn latent representations and try to align them using MMD. They test their approach across various domain shifts, including changes in observation, action, viewpoint, and agent morphology. Their proposed method is more stable and performant than adversarial approaches.

(b) Strengths
(+) (WwQw) Well-structured paper, reasonable approach: The paper proposes a series of simple, reasonable losses to align latent representations. The paper is also an easy read.
(+) (EC2z) Comprehensive evaluation: The paper evaluates their approach across many transfer scenarios and provides insightful ablations.

(c) Weaknesses
(-) (NwUM) Arbitrary transfer scenarios: The transfer tasks seem rather arbitrary. The authors seem to be cherry picking the tasks, and should consider all combinations of transfers to back up their claims.
(-) (NwUM) Simplistic robot experiments: The robot experiments are all in the end effector space with different viewpoints, rendering them far too simplistic to draw meaningful conclusions about the approach
(-) (EC2z, 9N3r, WwQw) Novelty: There's no theoretical foundation for the proposed method, nor where it would be performant and where it would be detrimental. Notably, MMD and TTC have been used in many transfer learning settings, which the paper fails to cite.
(-) (QnVM) Concern about experiment results: Proposed method appears to be not as robust (large std) and sensitive to hyperparameters

**Justification For Why Not Higher Score:**

The weaknesses of the paper outweigh the strengths. Notably, the lack of novelty, lack of analysis, and insufficient experiment evaluations.

**Justification For Why Not Lower Score:**

N/A

---

### Decision · Program_Chairs · 2024-01-16

Reject